# Measurement of Electrical Impedance Tomography-Based Regional Ventilation Delay for Individualized Titration of End-Expiratory Pressure

**DOI:** 10.3390/jcm10132933

**Published:** 2021-06-30

**Authors:** Thomas Muders, Benjamin Hentze, Stefan Kreyer, Karin Henriette Wodack, Steffen Leonhardt, Göran Hedenstierna, Hermann Wrigge, Christian Putensen

**Affiliations:** 1Department of Anesthesiology and Intensive Care Medicine, University Hospital Bonn, 53127 Bonn, Germany; benjamin.hentze@ukbonn.de (B.H.); Stefan.Kreyer@ukbonn.de (S.K.); Karin.Wodack@ukbonn.de (K.H.W.); Christian.Putensen@ukbonn.de (C.P.); 2Chair for Medical Information Technology, RWTH Aachen University, 52074 Aachen, Germany; leonhardt@hia.rwth-aachen.de; 3Department of Medical Sciences, Clinical Physiology, Uppsala University, 75185 Uppsala, Sweden; Goran.Hedenstierna@medsci.uu.se; 4Department of Anesthesiology, Intensive Care and Emergency Medicine, Pain Therapy, Bergmannstrost Hospital Halle, 06112 Halle, Germany; hermann.wrigge@bergmannstrost.de

**Keywords:** acute respiratory distress syndrome, positive end-expiratory pressure, individualized therapy, electrical impedance tomography, monitoring, functional imaging

## Abstract

Rationale: Individualized positive end-expiratory pressure (PEEP) titration might be beneficial in preventing tidal recruitment. To detect tidal recruitment by electrical impedance tomography (EIT), the time disparity between the regional ventilation curves (regional ventilation delay inhomogeneity [RVDI]) can be measured during controlled mechanical ventilation when applying a slow inflation of 12 mL/kg of body weight (BW). However, repeated large slow inflations may result in high end-inspiratory pressure (P_EI_), which might limit the clinical applicability of this method. We hypothesized that PEEP levels that minimize tidal recruitment can also be derived from EIT-based RVDI through the use of reduced slow inflation volumes. Methods: Decremental PEEP trials were performed in 15 lung-injured pigs. The PEEP level that minimized tidal recruitment was estimated from EIT-based RVDI measurement during slow inflations of 12, 9, 7.5, or 6 mL/kg BW. We compared RVDI and P_EI_ values resulting from different slow inflation volumes and estimated individualized PEEP levels. Results: RVDI values from slow inflations of 12 and 9 mL/kg BW showed excellent linear correlation (*R*^2^ = 0.87, *p* < 0.001). Correlations decreased for RVDI values from inflations of 7.5 (*R*^2^ = 0.68, *p* < 0.001) and 6 (*R*^2^ = 0.42, *p* < 0.001) mL/kg BW. Individualized PEEP levels estimated from 12 and 9 mL/kg BW were comparable (bias −0.3 cm H_2_O ± 1.2 cm H_2_O). Bias and scatter increased with further reduction in slow inflation volumes (for 7.5 mL/kg BW, bias 0 ± 3.2 cm H_2_O; for 6 mL/kg BW, bias 1.2 ± 4.0 cm H_2_O). P_EI_ resulting from 9 mL/kg BW inflations were comparable with P_EI_ during regular tidal volumes. Conclusions: PEEP titration to minimize tidal recruitment can be individualized according to EIT-based measurement of the time disparity of regional ventilation courses during slow inflations with low inflation volumes_._ This sufficiently decreases P_EI_ and may reduce potential clinical risks.

## 1. Introduction

In patients suffering from acute respiratory distress syndrome (ARDS), mechanical ventilation is necessary to ensure sufficient gas exchange; however, it may aggravate lung injury. Ventilator-induced lung injury (VILI) [1,2,3,4] may result from both end-inspiratory overdistension and tidal recruitment (cyclic opening of the collapsed lung tissue). Limiting tidal volume (V_T_) and end-inspiratory pressure (P_EI_) can reduce VILI and improve outcomes [5]. In contrast, the use of higher positive end-expiratory pressure (PEEP) levels does not reduce mortality rates among patients with ARDS [6,7]. Individual differences in the potential for alveolar recruitment might explain this situation [8]. Individualized PEEP settings aiming at improved and sustained lung recruitment [9] and at reduced tidal recruitment [10] may improve outcome in patients with ARDS who have recruitable lung collapse. Hence, detecting tidal recruitment might help optimize PEEP settings.

The global indices of lung mechanics do not enable the detection of regional collapse and tidal recruitment [11]. Electrical impedance tomography (EIT), in contrast, allows for the noninvasive monitoring of regional ventilation at the patient’s bedside [12,13,14]. We introduced an EIT-based parameter (regional ventilation delay inhomogeneity [RVDI]) to quantify the time disparity of the regional ventilation curves during slow inflation of 12 mL per kilogram of body weight (BW) [15,16]. RVDI enables the estimation of PEEP-associated changes in tidal recruitment [16] during controlled mechanical ventilation. Muders et al. recently demonstrated that a RVDI-based PEEP titration strategy minimizes tidal recruitment and improves regional ventilation/perfusion matching [17]. However, at higher PEEP levels, slow inflations of 12 mL/kg BW might potentially result in harmful increases in P_EI_. Therefore, the repeated use of these inflations during PEEP titration may contribute to VILI; thus, the clinical applicability of this approach is limited. In a previous study, Muders et al. demonstrated that RVDI can also be measured from slow inflations with a reduced inflation volume to quantify tidal recruitment while P_EI_ is limited [18]. The influence of this modified approach on derived PEEP levels, however, remains unclear.

We hypothesized that we could use EIT to identify PEEP levels that minimize the time disparity of the regional ventilation curves by measuring RVDI from slow inflations with reduced inflation volume during a decremental PEEP trial. To test this hypothesis, we performed extended analyses of data from an experimental model of lung injury.

## 2. Materials and Methods

### 2.1. Animals and Ethics

This research was based on an extended new analysis of raw data about PEEP titration from a recent study of lung injury in a swine model [17]. Experiments were approved by the Animal Research Ethics Committee of Uppsala University (C274/7) and performed in the Hedenstierna Laboratory, Department of Surgical Sciences, Uppsala University Hospital, Uppsala, Sweden, in accordance with the Guide for the Care and Use of Laboratory Animals (National Academy of Sciences, 1996) [19]) and the 3Rs principle. Reduction was achieved through a noninvasive technique for functional lung imaging (EIT). The animals were bought from a local farm and came from several long-standing colonies (Swedish landraces). The pigs had free access to feed and fresh water until 12 h before the start of the study. They were immediately transported to the laboratory before the start of the study. Proven medication regimens were used for analgesia, sedation, and euthanasia [16,17,18]. An adequate level of anesthesia was confirmed by paw clamping before continuous paralysis was started [16,17,18] (refinement). To allow for the complete replacement of additional experiments and ensure enhanced utilization of the previous study [17], this analysis was based on data simulations derived from already existing experimental animal data. Reporting of the study follows the Animal Research: Reporting of In Vivo Experiments (ARRIVE) guidelines [20].

### 2.2. Study Protocol

#### 2.2.1. Anesthesia, Animal Preparation, and Induction of Lung Injury

Premedication, aesthesia induction and maintenance, tracheotomy, and instrumentation were performed in 15 healthy pigs (9 males and 6 females), as previously described in detail [16,17,18]. Experimental lung injury was induced through a combination of elevated intra-abdominal pressure and titrated central venous injections of oleic acid in accordance with a well-established protocol [16,17,18].

#### 2.2.2. Baseline Ventilatory Setting

Pigs were mechanically ventilated to achieve a tidal volume (V_T_) of 8 mL/kg and a PEEP of 5 cm H_2_O by means of a volume-controlled mode. The fraction of inspired oxygen was initially 0.5 and was increased to 1.0 after the induction of lung injury [17].

#### 2.2.3. Lung Recruitment and Decremental PEEP Titration

To achieve maximal lung recruitment, we increased PEEP in a stepwise manner to 45 cm H_2_O, as reported elsewhere [17,21]. For PEEP titration, PEEP was decreased every 4 min in steps of 2 cm H_2_O, starting at 30 cm H_2_O. At each step of decrease, arterial blood gases were measured, and time disparities between the regional ventilation curves [17] were quantified during a single slow inflation with an inflation volume of 12 mL/kg BW using EIT to estimate tidal recruitment [17], as described in Section 2.3.3. The PEEP trial proceeded until a PEEP of 0 or a stop criterion (partial pressure of oxygen of <55 mm Hg, oxygen saturation of <88%, or mean arterial blood pressure of <55 mm Hg) was reached [17].

### 2.3. Measurements and Data Analysis

#### 2.3.1. Cardiovascular Measurements and Lung Mechanics

Details about cardiovascular, ventilatory, and lung mechanic measurement have been previously reported [17]. Respiratory curves were recorded and stored with the ventilator’s built-in monitoring system (Engström Carestation^TM^, GE Healthcare, Munich, Germany).

#### 2.3.2. Electrical Impedance Tomography

We used the EIT Evaluation Kit 2 (Dräger Medical GmbH, Lübeck, Germany) to create images of regional ventilation distribution. Impedance changes were measured and stored with a temporal resolution of 20 Hz and compared with a reference state. We used a modified Newton-Raphson reconstruction algorithm [16,17,18].

#### 2.3.3. Quantification of the Time Disparity of the Regional Ventilation Curves

Methods for RVDI determination have been previously described and extensively discussed in validation studies [16,18]. During slow inflation with a constant gas flow, regional ventilation delay (RVD) times were measured until the regional impedance time curve reached a threshold of 40% of the regional impedance maximum. Delay times were normalized to inflation time and expressed as RVD index (expressed as the percentage of inflation time). RVDs were plotted in a color-coded delay map (Figure 1). In contrast to a tidal image (Figure 1), which is a surrogate of the relative regional distribution of the tidal volume, the delay map illustrates the regional differences in ventilation timing. The time disparities between regional ventilation curves were quantified by RVDI, measured as standard deviations of all RVDs (Figure 1) [16,18]. RVDI is highly correlated with tidal recruitment [16,17,18].

RVDI was obtained from slow inflations with volumes of 12, 9, 7.5, and 6 mL/kg BW. To simulate slow inflation volumes (9, 7.5, and 6 mL/kg BW), we truncated the full EIT data recorded during the inflation of 12 mL/kg BW. Thus, we calculated RVDI at 100%, 75%, 62.5%, and 50% of inflation times, respectively, using a custom MATLAB software (MathWorks, Portola Valley, CA, USA), as previously described [18]. Additionally, RVDI was calculated from the regular tidal volumes during volume-controlled ventilation. Details are depicted in Figure 1.

#### 2.3.4. Calculation of P_EI_ from Reduced Slow Inflation Volumes

Using the SignalPlant software (Institute of Scientific Instruments, Czech Academy of Sciences, Staré Město, Czech Republic) [22], we assessed P_EI_ arising from reduced slow inflation volumes by truncating the recorded pressure signals. Thus, according to RVDI, pressure was calculated at 100%, 75%, 62.5%, and 50% of inflation times, as previously described [18].

#### 2.3.5. Selection of EIT-Based PEEP Levels from RVDI Values

Using EIT as described earlier, we derived RVDI from slow inflations and from regular V_T_ at all steps during the PEEP titration. Thus, for any PEEP step, RVDI was obtained from the applied slow inflation of 12 mL/kg BW, as well as from the simulated slow inflations of 9, 7.5, and 6 mL/kg BW, and from regular V_T_. By definition, EIT-based “individually optimized” PEEP level was the minimal PEEP level that prevented a progressive increase in RVDI, as previously described [17] (Figure 2).

### 2.4. Statistical Analysis

The primary outcome measures were EIT-based PEEP levels derived from RVDI measurements based on slow inflations with different inflation volumes during the decremental PEEP trial. Because of the lack of pilot or published data, sample size calculation was not feasible for this exploratory study setting.

End-inspiratory airway pressures associated with different slow inflation volumes during the decremental PEEP trial were compared in a two-way repeated-measures analysis of variance (ANOVA). If appropriate, consecutive post hoc tests (multiple comparison) were used to separate within-group differences. The derived EIT-based PEEP levels and RVDI values were compared in Bland–Altman analyses and linear correlation. All statistical analyses were performed with Prism 8 (GraphPad Software, San Diego, CA, USA).

## 3. Results

### 3.1. Availability of Data

No animal died before the whole study protocol was finished. Data of one animal (#2) had to be excluded because the EIT raw data were corrupted. Outlier data were not eliminated. The results from 14 animals were analyzed.

### 3.2. Cardiorespiratory Effects of Lung Injury Induction and Decremental PEEP Trial

Cardiorespiratory data are summarized in Table 1. The introduction of lung injury resulted in the impairment of oxygenation and lung mechanics that was comparable to moderate human ARDS [23]. Lung recruitment was successful in all animals. During the decremental PEEP trial, as PEEP decreased, P_EI_, and mean airway pressures also decreased. Driving pressure initially decreased with improvements in global respiratory compliance but increased when global respiratory compliance decreased at lower PEEP levels.

### 3.3. Temporal Heterogeneity Measured with EIT

Figure 3 shows interindividual linear correlations and the results of Bland–Altman analyses of RVDI values calculated from different slow inflation volumes. Very good linear correlation, low bias, and tight limits of agreement were found in the comparisons of RVDI values calculated from slow inflation volumes of 12 and 9 mL/kg BW (Figure 3A,E). Correlations decreased and bias and scatter increased when the slow inflation volume was further decreased to 7.5 (Figure 3B,F) and 6 (Figure 3C,G) mL/kg BW.

Bland-Altman analyses comparing RVDI values from slow inflations with an inflation volume of 12 mL/kg BW and RVDI values obtained from regular V_T_ showed high bias and low agreement (Figure 3H), and no significant linear correlation was found between these measures (Figure 3D).

In addition, linear correlations were calculated intraindividually for each animal (Table 2). In most of the pigs, correlations were excellent in the comparisons of RVDI values from slow inflations with an inflation volume of 9 vs. 12 mL/kg BW. Correlations fell with a further decline in slow inflation volumes. No significant intraindividual linear correlations were found when RVDI values from slow inflation volumes of 12 mL/kg BW were compared to RVDI values obtained from regular V_T_ (data not shown).

Intraindividual linear correlation analysis of regional ventilation delay inhomogeneity (RVDI; time disparities between regional impedance/time courses as the percentage of inflation time) calculated from different slow inflation volumes on electrical impedance tomography (EIT). RVDI from the simulated slow inflations of 9, 7.5, or 6 mL/kg of body weight (BW) was compared with RVDI from the original 12-mL/kg BW slow inflation during a decremental positive end-expiratory pressure (PEEP) trial. NA = not available.

The results of RVDI measured from different slow inflation volumes during a decremental PEEP trial are depicted in Figure 4. Temporal heterogeneity was low at higher PEEP levels and increased with decreasing PEEP. However, the curve progression of RVDI during the PEEP trial changed: RVDI during slow inflation volumes of 12 and 9 mL/kg BW showed a sigmoidal pattern (Figure 4B,C) with two inflection points. With lower slow inflation volumes (Figure 4C,D), RVDI appeared different without clear inflection points. RVDI calculated from regular V_T_ showed a different pattern. RVDI values were always low and did not exhibit a clear decrease or increase with changing PEEP levels (Figure 4F).

### 3.4. Individualized PEEP Levels Estimated from EIT

The individualized PEEP levels based on RVDI measurements calculated from different slow inflations during PEEP titration are depicted in Figure 5A. Figure 5B–D illustrates Bland–Altman analyses in which EIT-based PEEP levels estimated from slow inflation volumes of 9, 7.5, or 6 mL/kg BW were compared with those estimated from an slow inflation volume of 12 mL/kg BW. The reduction in slow inflation volume from 12 to 9 mL/kg BW resulted in exactly the same EIT-based PEEP in all except one animal. Hence, bias and scatter were low, and agreement was high (Figure 5B). With further reduction in slow inflation volume to 7.5 mL/kg BW, the mean EIT-based PEEP did not change (Figure 5A). However, Bland–Altman analysis revealed increased scatter and reduced agreement (Figure 5C). Measuring RVDI from the slow inflation volume of 6 mL/kg BW resulted in lower EIT-based PEEP levels (Figure 5A), the highest bias, and the lowest agreement (Figure 5D). EIT-based PEEP selection was not possible on the basis of RVDI values that were derived from regular V_T_.

### 3.5. P_EI_ Resulting from Different Slow Inflation Volumes during PEEP Titration

P_EI_ significantly decreased with decreasing PEEP during PEEP titration (Figure 6; in the ANOVA with the factor PEEP, *p* < 0.001). The reduction in slow inflation volumes resulted in significantly decreased P_EI_ at all PEEP steps (Figure 6; in the ANOVA with the factor inflation volume, *p* < 0.001, and in the ANOVA with the interaction PEEP * inflation volume, *p* = 0.0013). When compared with P_EI_ gained during “regular” ventilation with low V_T_, the P_EI_ levels during slow inflations of 12 mL/kg BW were significantly higher, whereas P_EI_ levels were the lowest during slow inflations of 6 mL/kg BW at higher PEEP levels (Figure 6; in post hoc multiple comparisons, *p* < 0.05). During the whole PEEP trial, P_EI_ levels resulting from slow inflation volumes of 9 and 7.5 mL/kg BW were comparable to P_EI_ levels resulting during ventilation with regular low V_T_ (Figure 6).

## 4. Discussion

The main results are as follows: RVDI values measured from slow inflation during a decremental PEEP trial depended on the inflation volume used for RVDI determination. In addition, the reduction in slow inflation volume sufficiently decreased the P_EI_ during the inflation. RVDI obtained from reduced slow inflation volumes but not from regular V_T_ during a decremental PEEP trial could be used to individualize PEEP so as to minimize tidal recruitment.

### 4.1. Influence of Slow Inflation Volume on RVDI, P_EI_, and Derived PEEP Levels during Decremental PEEP Trial

RVDI values (representing time disparities between regional ventilation/time courses) from slow inflations with a reduced inflation volume of 9 to 6 mL/kg BW are well correlated with RVDI values obtained from the original slow inflations of 12 mL/kg BW. This corroborates previous validation data [18].

Lower slow inflation volumes, however, resulted in lower RVDI values. These findings can be attributed to the underlying characteristics of RVDI, because RVDI represents the diversity of RVD times, which in turn depend on the total length of the slow inflation, as previously discussed [18]. During PEEP titration, this effect was more pronounced with lower slow inflation volumes and higher RVDI values (Figure 3F,G). Higher RVDI values, in turn, were associated with higher tidal recruitment [16,18], which was more frequently observed at lower PEEP levels in our lung injury model [16,17,18]. Thus, the detection of PEEP-related changes in tidal recruitment might be difficult when slow inflation volume is very low. The curve progression of the RVDI values during the decremental PEEP trial was clearly affected accordingly when the lowest slow inflation volumes (7.5 or 6 mL/kg BW) were used (Figure 4D,E), whereas the curve progressions of the RVDI values were similar when slow inflations of 12 or 9 mL/kg BW were used (Figure 4B,C). As a consequence, the choice of EIT-based PEEP levels (defined as the lowest PEEP level that avoided an progressive increase in RVDI) was affected when slow inflation volume was reduced to 7.5 or 6 mL/kg BW (Figure 5A,C,D). EIT-based PEEP levels differed when derived from slow inflations of 7.5 or 6 mL/kg BW (Figure 5C,D), whereas the derived EIT-based PEEP levels were comparable when RVDI was obtained from a slow inflation of 12 or 9 mL/kg BW (except in one animal, all derived PEEP levels were identical; Figure 5A,B).

RVDI values calculated from regular V_T_ did not correlate with RVDI values that were obtained from slow inflation volumes. Furthermore, individualized PEEP selection was not possible from the curve progression of V_T_-derived RVDI values (Figure 4F). As previously demonstrated [16]. RVDI measurements from regular V_T_ do not correlate with tidal recruitment. In our study, V_T_ was 8 mL/kg BW and thus the inflated volume was comparable to the lowest slow inflation volumes we used. However, inflation time of the regular V_T_ was substantially shorter when compared to slow inflation. This suggest that shortening the inflation time has a greater impact on RVDI calculation than the reduction in inflation volume. In our simulation study, we truncated the EIT raw data of the (constant flow) slow inflations, thus inflation time and inflation volume were reduced in equal measure. Therefore, we can only speculate that a PEEP titration would have yielded the same result if even smaller inflation volumes would have been applied, but with a slower gas flow and over a longer inflation period.

The use of high V_T_ may indeed be unsafe and result in critically high P_EI_ levels during mechanical ventilation. Slow inflations with even higher volumes and P_EI_, however, are widely and frequently applied to determine respiratory mechanics [24,25]. The effect of single slow-flow maneuvers allowing high inflation volume and P_EI_ remains unknown, and V_T_ or P_EI_ thresholds, which apply to mechanical ventilation, may not be fully valid for slow inflation maneuvers. However, when EIT-based PEEP is titrated, repeated slow inflations may be necessary to measure changes in RVDI. Moreover, repeated slow inflations might be required to adjust ventilatory settings over time. In a recent animal study [26], Haase et al. showed that our RVDI-based approach to titrate PEEP might slightly aggravate histopathological findings of lung damage, in comparison with an oxygenation-guided open-lung approach. Their findings were attributed to the repeated number of slow inflations with an inflation volume of 12 mL/kg BW during a 24-h experimental period [26]. Thus, the reduction in slow inflation volume to reduce the applied P_EI_ is clearly prudent. In this study, any decrease in slow inflation volume caused significant reduction in P_EI_ (Figure 6). In comparison with P_EI_ reached during regular tidal volumes (Figure 6, black symbols, connected by the dotted line), P_EI_ resulting from the 12 mL/kg BW slow inflation volumes were always higher at PEEP levels above 10 cm H_2_O. In contrast, P_EI_ did not significantly increase when inflation volumes of 9 mL/kg BW were applied (Figure 6).

In summary, our data showed that slow inflation volume can be reduced to 9 mL/kg BW to efficiently limit P_EI_, without changing the resulting EIT-based PEEP levels that were aimed at minimizing RVDI and hence tidal recruitment. Thus, a slow inflation volume of 9 mL/kg BW seems to be the best compromise.

### 4.2. Individualized PEEP Titration Based on Global and Regional Information about Lung Mechanics

Setting PEEP on the basis of global lung mechanics is a common practice [27,28]. PEEP titration guided by EIT may individually not reach the same level as might be found according to the measurements of global lung mechanics. RVDI does not correlate with global respiratory system compliance [16]. A comparison of RVDI and driving pressure, obtained from the present decremental PEEP trials has been recently presented and discussed in detail (see supplemental digital content 13 of reference [17]). Whereas RVDI and driving pressure were well correlated in single animals, they were not in others. Overall linear correlation of RVDI and driving pressure was low (R^2^ = 0.33) [17]. The two approaches demonstrated different patterns during the decremental PEEP titration and resulted in different individualized PEEP levels (no linear correlation, high bias, and low agreement) [17]. Thus, EIT-based RVDI measurements based on regional mechanical heterogeneity are not replaceable with conventional measurements of global lung mechanics such as driving pressure [17]. In a comparable study, Beda et al. [29] showed that regional data on pulmonary mechanics may deviate from global information.

Meanwhile, several EIT-based methods are available to optimize PEEP on an individual basis [12,13,14]. These approaches entail the use of information about the distribution or homogeneity of end-expiratory lung volume or V_T_, or regional information about lung recruitment, collapse, and overdistension [14]. However, all these approaches focus on analyzing the differences between end-inspiratory and end-expiratory EIT signals. The decisive feature of EIT, in contrast, is its high temporal resolution. This allows the temporal course of the ventilation to be analyzed. Thus, EIT provides detailed information about regional processes that occur between the end-expiration and the end-inspiration. We therefore developed RVDI [15,16] to quantify time disparities in regional ventilation. Previous research [16] showed that slow inflation with a constant gas flow is obligatory for sufficient RVDI calculation and the assessment of tidal recruitment. In a recent study in patients with ARDS, Becher et al. [30] used EIT to individually adjust PEEP and V_T_. Using a modified algorithm for RVDI measurements during a slow inflation of 12 mL/kg BW, Becher et al. demonstrated that time disparities in regional ventilation were reduced with EIT-based PEEP settings in comparison with clinical PEEP settings.

In our study, the absolute RVDI values were affected by PEEP (probably because of the PEEP-related extent of tidal recruitment) and slow inflation volume. Therefore, absolute RVDI values are not interchangeable when measured from different slow inflation volumes and inflation times. Comparing individual values of RVDI measurements obtained from different individuals may yield deceptive findings, but relative changes in RVDI values during PEEP titration might be more informative [18]. Minimal tidal recruitment might be indicated by the individual minimum of RVDI, allowing for the selection of the minimal PEEP step that minimizes tidal recruitment [16,18]. In the present and previous study [17] individualized EIT-based PEEP was defined as lowest PEEP level that avoids a progredient increase in RVDI. This criterion enables PEEP selection even if no clear minimum of RVDI can be identified from the RVDI curve progression. Nestler et al. [31] used this approach in patients with morbid obesity who had healthy lungs to individualize PEEP during general anesthesia. Muders et al. recently demonstrated that this RVDI-based PEEP titration strategy minimizes tidal recruitment and improves regional ventilation/perfusion matching in comparison with an Acute Respiratory Distress Syndrome Clinical Network (ARDSNet) protocol or an oxygenation-guided open lung strategy [17].

Our findings showed that comparable information could be obtained through the use of a reduced slow inflation volume, which reduced the applied P_EI_ to a safe level. Further studies are needed to explore whether individual PEEP titration based on EIT-derived measures of regional ventilatory heterogeneity is beneficial in patients with ARDS.

### 4.3. Limitations

Our study had clear limitations. ARDS subsumes multiple pathophysiologic phenotypes that cannot be imitated by a single experimental model. In our recruitable lung injury model [16,17,18,32], we combined oleic acid injection [33], which caused endothelial damage and edema [34,35,36], with elevated intra-abdominal pressure [37], which increased lung collapse and tidal recruitment [15,16]. Both are frequently observed in cases of extrapulmonary ARDS, such as those involving abdominal sepsis [38]. However, the results may not be applicable in patients with less recruitable ARDS (e.g., cases of pneumonia).

In our model, elevated intra-abdominal pressure caused relatively high P_EI_, inasmuch as chest wall compliance was impaired. We probably cannot prove that end-inspiratory transpulmonary pressures were lower because we did not measure esophageal pressures. Because intra-abdominal pressure affects transpulmonary pressure, our model may suggest that higher PEEP values are preferable and that individualized PEEP levels might deviate as a result of other intra-abdominal pressure values [17]. Our EIT-based method to individually titrate PEEP aims at minimizing tidal recruitment [16,17,18,26]. Its ability to detect inspiratory overdistension is not validated. However, Muders et al. recently demonstrated that the use of our method also leads to a decrease in dead space ventilation, which might indicate a reduction in overdistension [17].

A P_EI_ of up to 50 cm H_2_O was necessary to provide complete lung recruitment in our animal model; otherwise, lung volume history might have influenced our measurements. We are aware that recruitment can be deleterious, particularly in patients with nonrecruitable lungs, and we do not advocate these maneuvers as routine procedures in all patients undergoing ventilation.

Measurement of RVDI requires controlled mechanical ventilation and the application of a constant gas flow. Thus, the approach is not applicable during spontaneous breathing activity. There is no evidence that the individualized PEEP titration presented here improves patient outcome. Therefore, we cannot recommend once per day RVDI measurement if sedation of the patient is necessary to allow the measurement to be taken.

To simulate reduced slow inflation volumes, we truncated inflation time in the EIT raw data. These simulations were based on the assumption that, with constant gas flow during slow inflation, inflation volume will proportionally decrease when inflation time decreases. Muders et al. used this simulation algorithm in a previous validation study [18], demonstrating that the RVDI measurements obtained from (simulated) reduced slow inflation volumes were well correlated with tidal recruitment as measured from computed tomography. In accordance with the 3Rs principle, data simulation enabled the extended use of existing experimental animal data.

## 5. Conclusions

In our animal model, we performed individualized EIT-based PEEP titration aimed at minimizing RVDI, and hence tidal recruitment, by measuring time disparities of regional ventilatory courses during slow inflation with a low inflation volume_._ The reduction in slow inflation volume sufficiently decreases P_EI_. This might diminish the potential risks in the clinical implementation of this approach. Further studies are needed to verify our results in patients with ARDS.

## Figures and Tables

**Figure 1 jcm-10-02933-f001:**
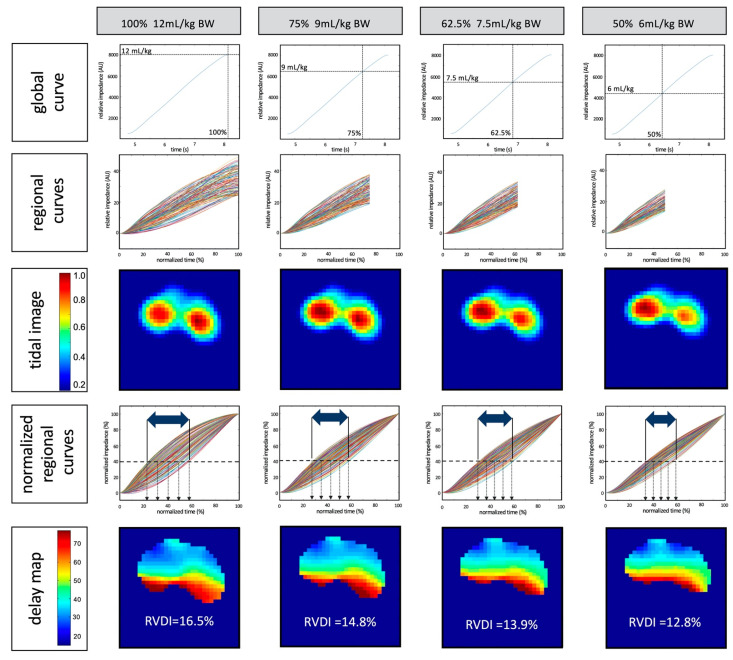
Calculation of regional ventilation delay inhomogeneity (RVDI) to quantify the time disparities between regional ventilation curves that are correlated with tidal recruitment [16]. Left column: calculation from the original 12 mL/kg of body weight (BW; 100%) slow inflation; other columns (left to right): calculation from the simulated reduced inflation volume of 9, 7.5, and 6 mL/kg BW at 75%, 62.5%, and 50% inflation times, respectively. Rows (top to bottom): global impedance/time curves; regional impedance/time curves; resulting electrical impedance tomography (EIT) images of tidal ventilation; normalized regional impedance/time curves, in which downward arrows indicate regional ventilation delay (RVD) indices (delay times at which regional curves reach the regional 40% inflation threshold and horizontal two-sided arrows symbolize time disparity); and delay maps with resulting RVDI values. Methodological schematic description, adapted from Muders et al. [18].

**Figure 2 jcm-10-02933-f002:**
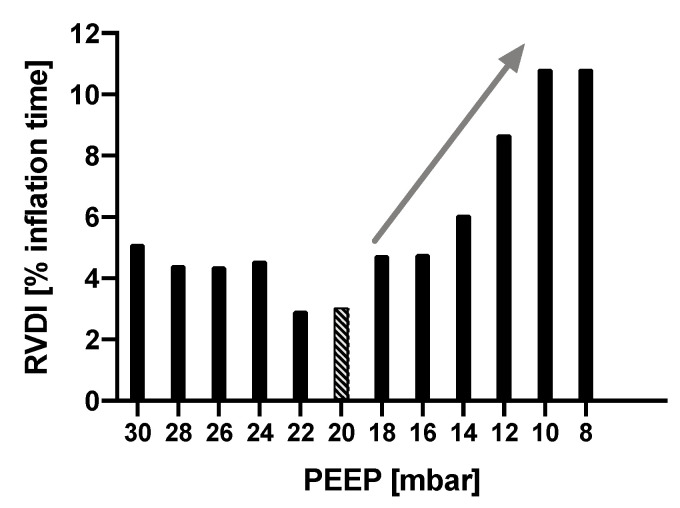
Measurements of time disparities (regional ventilation delay inhomogeneity [RVDI]) derived from slow inflations during a decremental positive end-expiratory pressure (PEEP) trial. RVDI is minimized at a PEEP of 20 cm H_2_O. Below this PEEP, RVDI progressively increased (see arrow), which suggested that tidal recruitment also increased [16,18]. Thus, a PEEP of 20 cm H_2_O was presumably associated with minimal tidal recruitment [17].

**Figure 3 jcm-10-02933-f003:**
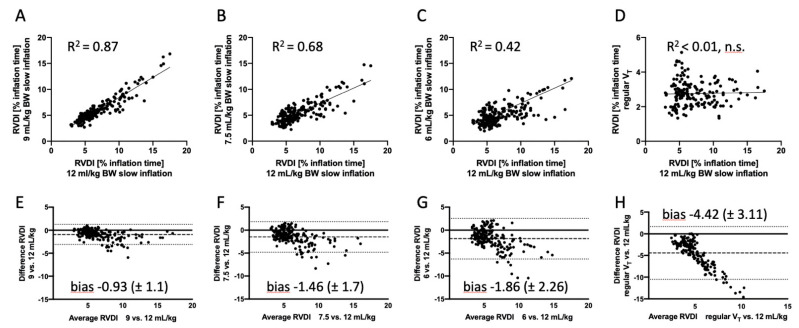
(**A**–**D**) Comparison (linear correlation) of regional ventilation delay inhomogeneity (RVDI) values (time disparity of regional impedance/time courses as the percentage of inflation time) calculated from different slow inflation volumes and V_T_ depicted on electrical impedance tomography (EIT). The x-axis represents RVDI from slow inflation with 12 mL/kg of body weight (BW); the y-axis represents RVDI from the simulated slow inflations of 9 (**A**), 7.5 (**B**), and 6 (**C**) mL/kg BW, and regular tidal volume (V_T_) (**D**). (**E**–**H**) Comparison of means with differences between RVDI from the original slow inflation of 12 mL/kg BW and RVDI from the simulated slow inflations of 9 (**E**), 7.5 (**F**), and 6 (**G**) mL/kg BW, and regular V_T_ (**H**). Bias ± standard deviations of bias are shown within the graphs. Limits of agreement are indicated by the dotted lines.

**Figure 4 jcm-10-02933-f004:**
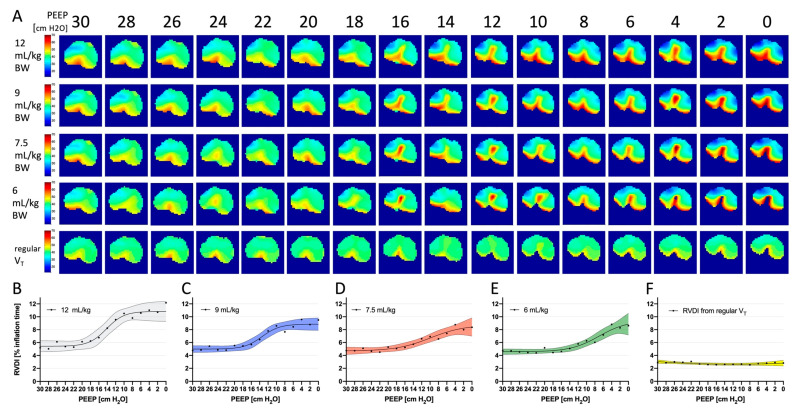
(**A**) Time disparities between regional impedance/time courses, displayed as color-coded regional ventilation delay (RVD) maps (as the percentage of inflation time) and calculated from slow inflation volumes of 12, 9, 7.5, and 6 mL/kg of body weight (BW), and regular tidal volume (V_T_) on electrical impedance tomography (EIT); decremental positive end-expiratory pressure (PEEP) titration (30 to 0 cm H_2_O). Data of a representative animal. (**B**–**F**) Measures of regional ventilation delay inhomogeneity (RVDI; quantifying the inhomogeneity of the RVD maps) during the decremental PEEP trial obtained from 12 (gray), 9 (blue), 7.5 (red), and 6 (green) mL/kg, and regular V_T_ (yellow). Black dots indicate means. Solid lines indicate a fitted polynomial curve of means. Dotted lines and colored areas demonstrate the 95% confidence interval. Note: *x*-axis shows decreasing PEEP values from 30 to 0 cm H_2_O according to the decremental trial.

**Figure 5 jcm-10-02933-f005:**
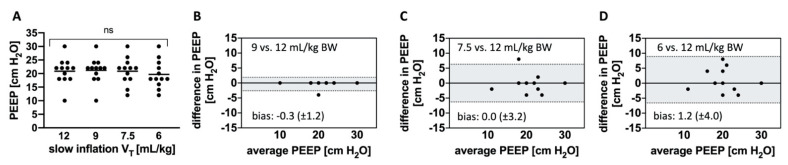
(**A**) Electrical impedance tomography (EIT)-based positive end-expiratory pressure (PEEP) levels derived from EIT-based regional ventilation delay measurements from different slow inflation tidal volume (V_T_) during a decremental PEEP trial. Dots indicate individual PEEP levels. Solid lines show means. ns = not significant, repeated-measures analysis of variance (ANOVA). (**B**–**D**) Bland–Altman comparisons of EIT-based PEEP levels derived from (**B**) 9, (**C**) 7.5, and (**D**) 6 mL/kg of body weight (BW) with the original 12 mL/kg BW slow inflation. Bias ± standard deviations of bias are shown within the graphs. Limits of agreement are indicated by the dotted lines enclosing the gray areas.

**Figure 6 jcm-10-02933-f006:**
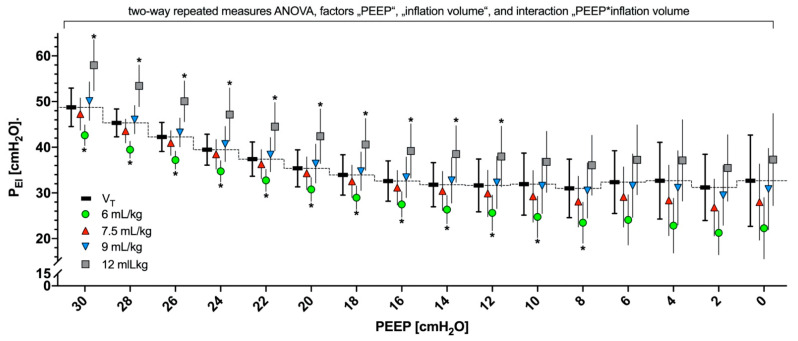
End-inspiratory pressure (PEI) resulting from slow inflation with tidal volumes (VTs) of 12 (gray), 9 (blue), 7.5 (red), and 6 (green) mL/kg and regular VTs of 8 mL/kg of body weight (BW; black, connected by the dashed line) during a decremental PEEP trial from 30 to 0 cm H_2_O. Repeated-measures analysis of variance (ANOVA) included the factors PEEP, inflation volume, and the interaction of these factors. * Asterisks indicate the significant results of consecutive post hoc tests (*p* < 0.05, multiple comparison) for the differences between PEI resulting from different slow inflation volumes and PEI resulting from regular low VTs.

**Table 1 jcm-10-02933-t001:** Cardiorespiratory effects of lung injury induction and decremental PEEP trial.

Measure	[Unit]	Baseline	after ALI	PEEP Titration															
PEEP	[cm H_2_O]	5	9 ± 2	30	28	26	24	22	20	18	16	14	12	10	8	6	4	2	0
dP_aw_	[cm H_2_O]	14 ± 3	27 ± 7	16 ± 5	15 ± 3	14 ± 3	13 ± 4	13 ± 4	13 ± 4	13 ± 5	14 ± 4	15 ± 5	17 ± 6	19 ± 7	20 ± 8	22 ± 7	25 ± 9	27 ± 7	30 ± 10
RR	[L/min]	27 ± 3	33 ± 4	35 ± 1	35 ± 1	35 ± 1	35 ± 1	35 ± 1	35 ± 1	35 ± 1	35 ± 1	35 ± 1	36 ± 1	36 ± 2	36 ± 2	36 ± 2	35 ± 0	35 ± 0	35 ± 0
V_T_	[ml/kg BW]	8.1 ± 0.8	8.1 ± 0.7	8.1 ± 0.8	8.0 ± 0.8	8.1 ± 0.7	7.9 ± 0.8	8.1 ± 0.7	8.0 ± 0.7	7.9 ± 0.9	7.9 ± 0.8	8.0 ± 0.8	8.1 ± 0.8	8.0 ± 0.8	7.9 ± 0.9	8.0 ± 0.8	8.1 ± 0.8	8.1 ± 0.8	8.0 ± 0.7
HR	[L/min]	112 ± 17	117 ± 11	126 ± 35	122 ± 33	119 ± 31	116 ± 28	114 ± 27	112 ± 25	112 ± 24	111 ± 23	112 ± 23	112 ± 22	114 ± 23	112 ± 19	116 ± 22	109 ± 12	108 ± 6	110 ± 6
MAP	[mm Hg]	93 ± 10	107 ± 16	97 ± 18	98 ± 17	99 ± 16	100 ± 18	102 ± 15	103 ± 15	104 ± 15	104 ± 14	103 ± 14	104 ± 14	104 ± 14	106 ± 13	111 ± 14	111 ± 12	111 ± 11	109 ± 14
CVP	[mm Hg]	9 ± 3	13 ± 3	22 ± 3	21 ± 4	21 ± 3	20 ± 3	19 ± 3	18 ± 3	17 ± 3	17 ± 3	16 ± 3	15 ± 3	15 ± 3	14 ± 3	14 ± 3	12 ± 6	11 ± 6	10 ± 7
CO	[L/min]	4.7 ± 0.8	5.3 ± 1.3	5.0 ± 1.4	4.9 ± 1.4	4.9 ± 1.5	4.9 ± 1.4	5.1 ± 1.5	5.2 ± 1.6	5.3 ± 1.5	5.3 ± 1.5	5.4 ± 1.6	5.5 ± 1.6	5.5 ± 1.6	5.5 ± 1.5	5.7 ± 1.5	6.3 ± 1.9	6.3 ± 2.3	5.2 ± 2.3
F_i_O_2_		0.5 + 0	0.5 + 0.1	1.0 + 0	1.0 + 0	1.0 + 0	1.0 + 0	1.0 + 0	1.0 + 0	1.0 + 0	1.0 + 0	1.0 + 0	1.0 + 0	1.0 + 0	1.0 + 0	1.0 + 0	1.0 + 0	1.0 + 0	1.0 + 0
P_a_O_2_	[mm Hg]	215 ± 37	92 ± 18	465 ± 128	437 ± 170	467 ± 143	445 ± 150	434 ± 149	409 ± 151	365 ± 152	309 ± 144	262 ± 141	210 ± 137	165 ± 127	151 ± 103	129 ± 82	135 ± 71	118 ± 56	115 ± 54
*n*		14	14	14	14	14	14	14	14	14	14	14	13	13	11	10	6	5	3

PEEP: positive end-expiratory pressure, dP_aw_: driving pressure, RR: respiratory rate, V_T_: tidal volume, MAP: mean arterial blood pressure, CVP: central venous blood pressure, CO: cardiac output, F_i_O_2_: fraction of inspired oxygen, P_a_O_2_: arterial oxygen partial pressure, n: number of animals finishing the PEEP step.

**Table 2 jcm-10-02933-t002:** Intraindividual linear correlation analysis of RVDI.

Slow Inflation Volume [mL/kg BW]	Linear Correlation, *R*^2^, *p* < 0.001, Respectively														
	Pig Number														
	1	2	3	4	5	6	7	8	9	10	11	12	13	14	15
9 vs. 12	0.95	NA	0.42	0.89	0.96	0.98	0.96	0.95	0.93	0.95	0.29	0.76	0.90	0.91	0.92
7.5 vs. 12	0.91	NA	0.22	0.84	0.92	0.98	0.91	0.89	0.76	0.82	0.11	0.43	0.66	0.45	0.86
6 vs. 12	0.91	NA	0.27	0.59	0.77	0.97	0.74	0.84	0.63	0.62	0.30	0.07	0.01	0.04	0.80

## Data Availability

The authors confirm that the data supporting the findings of this study are available within the article.

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
