# Peer review of "Measurement of Electrical Impedance Tomography-Based Regional Ventilation Delay for Individualized Titration of End-Expiratory Pressure"

_jcm, 2021, doi:10.3390/jcm10132933_

Round 1
Reviewer 1 Report
In their study, Muders and Coworkers compared different simulated tidal volumes during slow inflation maneuvers for selection of PEEP levels with RVDI. The manuscript is well written and addresses an important question. I have the following comments:
Major comments:
RVDI is clinically available not only for analyzing sustained-inflation maneuvers, but also for analyzing normal tidal ventilation. In daily clinical practice, clinicians might be tempted to perform these analyses during normal tidal ventilation, omitting the cumbersome sustained inflation maneuvers. However, it is highly questionable whether analyzing normal tidal ventilation will yield similar results. For comparison, could you please report the RVDI values and EIT-based PEEP levels that would results if the analysis was performed during normal tidal ventilation? This would be important information to guide clinical practice.
Minor comments:
There are some minor typing errors that should be corrected before print (e.g., line 64: “ventilation curvescan” or (line 97) “lung injury induction.,.” or (line 104) “PEEP trial was proceded” or (line 137) “Thus, , according” and many others)
Introduction: I am not sure if the statement “Individualized PEEP setting aiming at improved and sustained lung recruitment and reduced tidal recruitment has been shown to improve outcome in ARDS patients with recruitable lung collapse” is correct. To my knowledge, there has been no prospective randomized study showing improved outcomes with individualized PEEP. If there is one, please cite it. Otherwise, I suggest rephrasing to “Individualized PEEP setting aiming at improved and sustained lung recruitment and reduced tidal recruitment may improve outcome in ARDS patients with recruitable lung collapse.”
Unfortunately I was unable to find table S1, which was supposed to be in an online supplement. I think information on basic cardiorespiratory parameters like changes in compliance and driving pressure during the PEEP trial (but also mean arterial pressure and central venous pressure) should not be hidden in an online supplement but should be presented in the main manuscript in table 1. As there is currently only one table in the main manuscript, I suggest removing the online supplement, making what is now table S1 the new table 1 (table 1 would then become table 2).
It would also be interesting to see if there was any correlation between RVDI and global compliance / driving pressure as well as RVDI-selected PEEP and PEEP selected according to driving pressure / global Crs
Reference 18 is incomplete (no journal / volume is given)
Chapter 3.4: How exactly were the “individualized PEEP levels estimated from EIT” determined? Figure 2 suggests that there was a clear minimum for RVDI, but looking at the time courses with VT 7.5 and VT 6 (Fig 4), I imagine that in some cases it might be difficult to determine the exact PEEP level with minimal RVDI or that there may have been some ambiguity. How was this resolved?
How high was the gas flow during the slow inflation maneuver? Was the gas flow constant for VT 12, 9, 7.5 and 6 or was inflation time constant? Could this have an influence on the results, explaining the greater scatter at VT 7.5 and 6?
In the end, you demonstrated that some reduction in VT during slow inflation is possible. However, it remains unclear which VT should be recommended. 9 ml? 7.5 ml? What is the best compromise? Please comment and discuss.
Reviewer 2 Report
In this manuscript, Muders et al. present a secondary analysis of an EIT-based approach for the setting of PEEP during mechanical ventilation.
The authors (experts in the filed) have to be congratulated for the study and for the very nice manuscript!
I have only major comment:
- Not all readers may be aware, that the analysis of RVDI is only possible while patients or animals are on controlled mechanical ventilation (i. e. no spontaneous breathing activity). Please insert this in the abstract and at least in the introduction section of the manuscript. Please discuss this limitation also in the discussion section. Would you paralyze (or deep sedate) your patients once per day to perform RVDI measurement?
Reference 29 has now been published:
Becher T, Buchholz V, Hassel D, Meinel T, Schädler D, Frerichs I, Weiler N.
Individualization of PEEP and tidal volume in ARDS patients with electrical impedance tomography: a pilot feasibility study.
Ann Intensive Care 89: https://doi.org/10.1186/s13613-021-00877-7, 2021.
Author Response
please see the attachment

This manuscript is a resubmission of an earlier submission. The following is a list of the peer review reports and author responses from that submission.